# Genome-wide scan for selection signatures in Mexican Sardo Negro Zebu cattle

Victor Isaias Garduño López[1], Ricardo Martínez-Rocha[1]*, Rafael Núñez Domínguez[2], Rodolfo Ramírez Valverde[2], Joel Domínguez Viveros[3], Antonio Reyes Ceron[4], Jorge Hidalgo[5]

1 Facultad de Estudios Superiores Cuautitlán, Universidad Nacional Autónoma de México, Cuautitlán Izcalli, Estado de México, México, 2 Posgrado en Producción Animal, Universidad Autónoma Chapingo, Texcoco, Estado de México, México, 3 Facultad de Zootecnia y Ecología, Universidad Autónoma de Chihuahua, Chihuahua, México, 4 Asociación Mexicana de Criadores de Cebú, Tampico, Tamaulipas, México, 5 Department of Animal and Dairy Science, University of Georgia, Athens, Georgia, United States of America

* ricardomtzrocha@cuautitlan.unam.mx

**Data Availability Statement:** The data that support the findings of this study are available in a public

## Abstract

The Sardo Negro cattle (SN) is the only zebu cattle breed developed in Mexico. Since its development, the selection could have led to an increase in the homozygosity level in some regions of the genome and made differentiation with other cattle populations. We aimed to identify and characterize selection signatures in SN using medium-density SNP data using four approaches: 1) Runs of homozygosity (ROH) 2) Nucleotide Diversity 3) Tajima's D and 4) the Wright's fixation index ($F_{ST}$). A sample of 555 SN animals genotyped for 65k SNPs was used to obtain ROH segments considered regions under selection. The $F_{ST}$ values were estimated by comparing the sample of genotyped SN animals with samples of genotyped animals from the Gir, Brahman, and Ongole breeds. Only one region mapped to 35.78–42.51 Mb on BTA6 was considered a selection signature by the ROH method. This selection signature overlapped with the lowest diversity, negative values of Tajima's D and a diversification region between SN and the other Zebu breeds by $F_{ST}$. We found several candidate genes (*LCORL*, *NCAPG*, and *SLIT2*) related to growth and other economically important productive traits in this common region. Using the $F_{ST}$ method, different regions, such as regions on BTA8 (8:93.4–93.9 Mb), BTA11 (11:99.2–99.7), and BTA14 (14: 26.1–26.8) related to growth and milk traits also were defined as candidate selection signatures. The selective signals identified in this study reflected the direction of the selection pressure that primarily involves the increase of live weight traits in the Sardo Negro cattle breeding program.

## Introduction

The Sardo Negro Zebu breed (SN) has been developed as a dual-purpose breed, i.e., the final target is to breed animals able to produce weaned calves with high growth potential and able to produce a good amount of milk. Because the SN animals exhibit good growth potential and milk production combined with superior heat tolerance, this cattle breed is mainly used in production systems in tropical regions of Mexico. It already has ~50,000 animals registered in its pedigree [1]. This breed was developed in the 1950s using Gir and Indubrasil bulls,

repository (https://doi.org/10.17632/pcytgvc7m3.1).

**Funding:** The author(s) received no specific funding for this work.

**Competing interests:** The authors have declared that no competing interests exist.

obtaining black and white fur animals, and since then, the genealogy of this breed has started to be recorded [2]. Until now, SN has been breeding independently of other Zebu breeds and with specific breeding goals; therefore, it is considered a new Mexican Zebu breed.

During the years of the SN establishment as a new breed, selecting specific phenotypes for *inter se* matings could shape the pattern of genetic variation so that some allele frequencies tend to change in a particular direction, and even allelic fixation could occur. Saravanan [3] mentioned that selection strategies reduce variation in specific genomic regions that control the breed characteristics such as morphology, body conformation, production, etc. Selection signatures are considered regions of the genome where genetic variability has diminished throughout generations due to selection practices. When directional selection leads to the fixation at some loci, other alleles in linkage disequilibrium also increase their frequencies, a phenomenon known as hitchhiking [4]. When this phenomenon favors retaining the favorable alleles, these genome regions are considered selective sweeps [5]. Selective sweeps are a process by which a new advantageous mutation eliminates or reduces variation in linked neutral sites as it increases in frequency.

The methods for detecting selection signatures based on SNP genotyping technologies can be broadly classified into two groups: intra-population and inter-population statistics [3]. Based on the hitchhiking theory, the selective sweeps should have stretches of homozygous loci that exhibit higher homozygosity than the average of the genome; therefore, runs of homozygosity (ROH) can be used to identify selection signatures [6]. Wright's Fixation index ($F_{ST}$) [7] is based on differences in allele frequencies between populations; highly differentiated allele frequencies at any given locus between two populations indicate positive selection. We aimed to identify and characterize selection signatures in the SN cattle population using medium-density SNP data using two approaches: ROH and $F_{ST}$.

## Materials and methods

### Ethics statement

This study involved the analysis of pre-collected hair samples from animals and did not include any direct interaction with live animals. According to the Canadian Council on Animal Care (CCAC) guidelines, specifically the conditions outlined in the "Guide to the Care and Use of Experimental Animals", research using non-invasive methods and pre-existing samples do not require formal ethical review. As this research did not involve new procedures or the handling of live animals, and adhered to the ethical standards in place at the time of sample collection, it qualifies for exemption criteria.

### Animals and sampling

We use hair follicle samples from the SN, which were pre-collected by the Asociación Mexicana de Criadores de Cebú (AMCC). The sampled SN animals included 40 males and 515 females born between 2013 and 2019 in six farms in Mexico's tropical areas. The study samples were randomly selected, including only animals with no parent-offspring relationship. Only animals were considered if they were purebred based on pedigree and farmer interviews. The hair samples were submitted for genotyping with a 65k SNP chip (Axiom Bovine Genotyping v3 Array, aligned to the *Bos taurus* genome assembly ARS-UCD1.2).

### Genotyping data

In this study, we used five sets of SNPs, i.e., genotypes from 1) 555 SN bovines; 2) 97 Mexican Brahman Zebu bovines (MxBr); 3) 20 Gir Zebu bovines; 4) 20 American Brahman Zebu bovines (Brah); and 5) 20 Ongole Zebu bovines (Ong).

Quality controls on animals and markers of the first two sets were performed using the PLINK software version 1.90 [8] according to the following parameters: minimum SNP call rate of 90%, minor allele frequencies > 0.01, only autosomal SNPs were considered, and only individuals with high genotyping (at least 90% complete call rate) were included; 35,321 SNPs and the 555 animals passed filters and quality control for SN and 35,294 SNPs and 97 animals for MxBr. The last three data sets were taken from a public repository [9]. These animals were genotyped with the Illumina Bovine SNP50 BeadChip array developed using UMD3.1 assembly. The authors already had done quality control for these datasets with the same parameters considered in this study. Only SN animal genotypes were considered for the selection signatures identification with ROH. For the identification of selection signatures using the $F_{ST}$ methods, the SN genotypes were merged with the other genotypes of each breed at a time. We identified the common genetic markers mapping SNPs to a common assembly using SNPchiMp database [10]. The representation an strand orientations were checked using the—flip and—flip-scan functions of PLINK. Finally, the combined datasets included SN-MxBr = 33,924 SNPs, SN-Gir = 18,410 SNPs, SN-Brah = 18,464 SNPs, and SN-Ong = 18,460 SNPs.

## Selection signatures intra-population

We calculated the ROH using the 'detectRUNS' R package [11], which implements a sliding-window-based method similar to that implemented in the PLINK software. The ROHs were calculated separately for every animal with the following parameters: (1) the minimum length that constituted the ROH was 1 Mb, (2) a minimum number of 20 consecutive homozygous SNPs was required to call a segment an ROH, and a maximum of 1 heterozygous SNPs was allowed in the homozygous SNP windows, (3) the maximum gap between consecutive SNPs was set to 1000 kb, and (4) the maximum number of missing allowed genotypes was set to 1. The minimum density considered and the maximum gap between consecutive SNPs were 1 SNP every 1000 kb and 1 Mb, respectively. The ROHs were classified into four categories according to length: 1 to < 2 Mb, 2 to < 4 Mb, 4 to < 8 Mb, and > 8 Mb.

The idea of detecting selection signatures by ROH relies on searching for chunks of the genome without heterozygosity in the diploid state on a genome-wide scale to identify signals of past selection [12]. In our study, the percentage of SNP existing within an ROH was estimated by counting the number of times each SNP appeared in an ROH and dividing that number by the number of animals [13]. Shared ROHs within a livestock population identify chromosome regions in which a reduced haplotype variability produces ROH islands. ROH islands were identified as a percentile threshold of the frequency of an SNP in ROH calculated within a breed (called percentile-based threshold), the threshold used in this work was the top 0.1% of SNPs with the highest occurrences [14,15].

Both the nucleotide diversity and Tajima's D were estimated based on a sliding window approach with windows of 50 kb and a step of 50 kb using VCFtools [16].

## Selection signatures inter-population

The $F_{ST}$ was estimated using pairwise breeds, i.e., considering Sardo Negro, and the other breeds as control. The $F_{ST}$ was calculated using the—Fst function of the PLINK software, which calculates the $F_{ST}$ for each autosomal diploid variant using the method introduced by Weir and Cockerham [17]. Excessively high $F_{ST}$ values at any given locus indicate a greater divergent selection between the populations than expected under drift. In contrast, excessively low $F_{ST}$ values indicate less divergence than expected under drift and subtle balancing selection between the tested populations. The regions with the top 0.1% $F_{ST}$ values and the SNP with

higher $F_{ST}$ values in more than three pairs of cattle breeds were considered to represent the signatures of selection. For each $F_{ST}$ peak, a window of 500 Kb (250 Kb upstream and 250 Kb downstream) was evaluated for gene annotation.

## Bioinformatic analyses

The identified selection signatures were compared for genes annotated to the *Bos taurus* genome assembly ASR-UCD2.0 using Genome Data Viewer (NCBI; https://www.ncbi.nlm. nih.gov/datasets/genome/GCA_002263795.4). Gene ontology (GO) significantly overrepresented terms were identified using PANTHER 18.0 (http://www.pantherdb.org/) [18].

## Results

In total, 2,674 ROH were identified among the 555 SN samples. Descriptive statistics of each length category are given in Table 1. Our results show that 21% of the ROH were shorter than 4 Mb. Most of the ROH for SN cattle population comprised segments longer than 4 Mb, i.e., segments 4–8 Mb and >8 Mb represented 83%. In general, animals with a larger ROH number tend to have a greater total length of ROH segments regardless of the length of single ROH regions (Fig 1).

A Manhattan plot shows the incidence of each SNP in the ROH across individuals (Fig 2A). The genomic regions most associated with ROH in all individuals were defined by the top 0.1% of SNPs observed in the ROH. The adjacent SNPs over this threshold were merged into genomic regions corresponding to a ROH island (Table 2).

We applied the nucleotide diversity analysis and Tajima's D method to detect genomic regions related to selection in the Sardo Negro breed (Fig 2B and 2C). Two methods showed outlier signals (lowest 1% with nucleotide diversity analysis; values under zero with Tajima's D) in overlapping regions and were therefore considered candidate selective regions. Our results showed that in the three intra-population methods, the selection signature on BTA6 is confirmed.

The identification of only one ROH island on BTA 6 (37.39–39.57 Mb) was the result of this approach. The annotation analyses revealed 25 candidate genes overlapping with that genomic region under positive selection detected using the $F_{ST}$ metric. Several QTLs have been reported in that genomic region. Most of the traits associated with those QTLs are related to growth.

Wright's $F_{ST}$ was calculated pairwise for each SNP in each group of animals to examine differentiation among Sardo Negro and the other Zebu breeds to determine signals of diversifying selection. The regions above 0.1% of the highest $F_{ST}$ values were considered signals of positive selection (S1 Table). We identified 80 genomic regions (50 unique regions) under divergent selection. The genome-wide distribution of selection signatures by $F_{ST}$ was plotted for the four breed pairs (S1–S4 Figs) and jointly (Fig 3). Nine SNP above the threshold coincided in the three comparisons with Sardo Negro (S2 Table).

**Table 1. Number, mean length, and genome coverage of different length classes runs of homozygosity (ROH).**

| Class | No of ROH | Percent | ROH mean | Genome coverage (%) |
|---|---|---|---|---|
| 1–2 Mb | 82 | 0.03 | 1.43 | 0.06 |
| 2–4 Mb | 474 | 0.18 | 3.33 | 0.13 |
| 4–8 Mb | 1187 | 0.44 | 5.61 | 0.22 |
| >8 Mb | 931 | 0.35 | 15.23 | 0.61 |
| Total | 2674 | 1.0 | 8.43 | 1.73 |

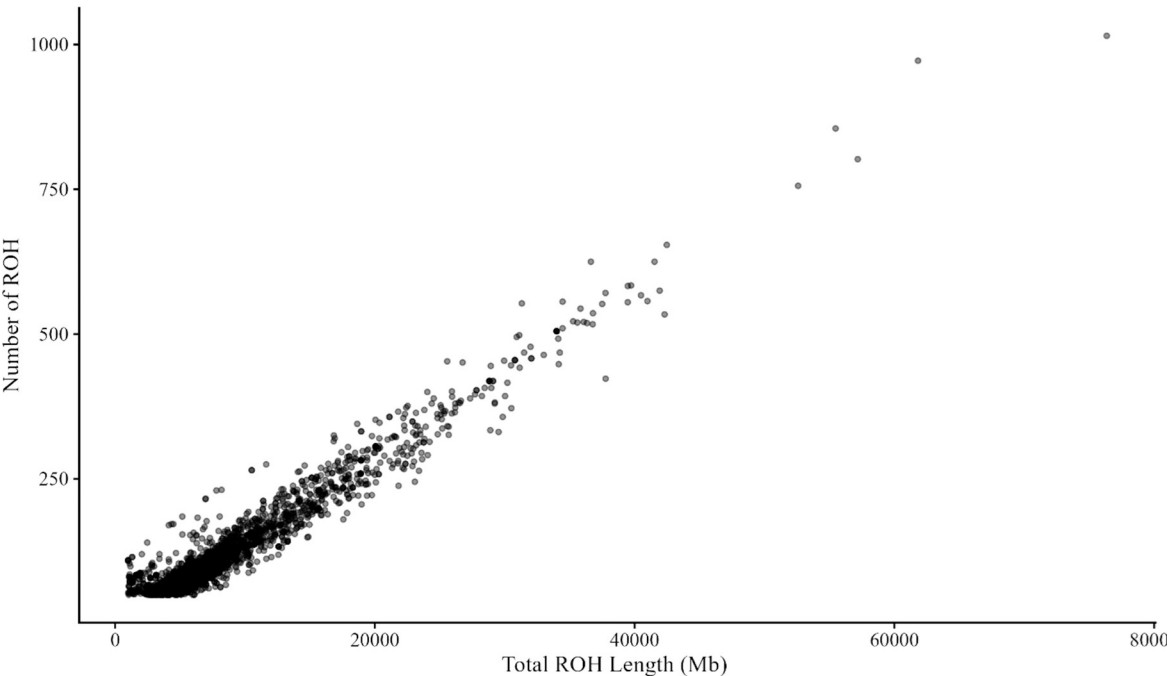

**Fig 1. Relationship between the number of runs of homozygosity (ROH) found in each individual and their total length (Mb).**

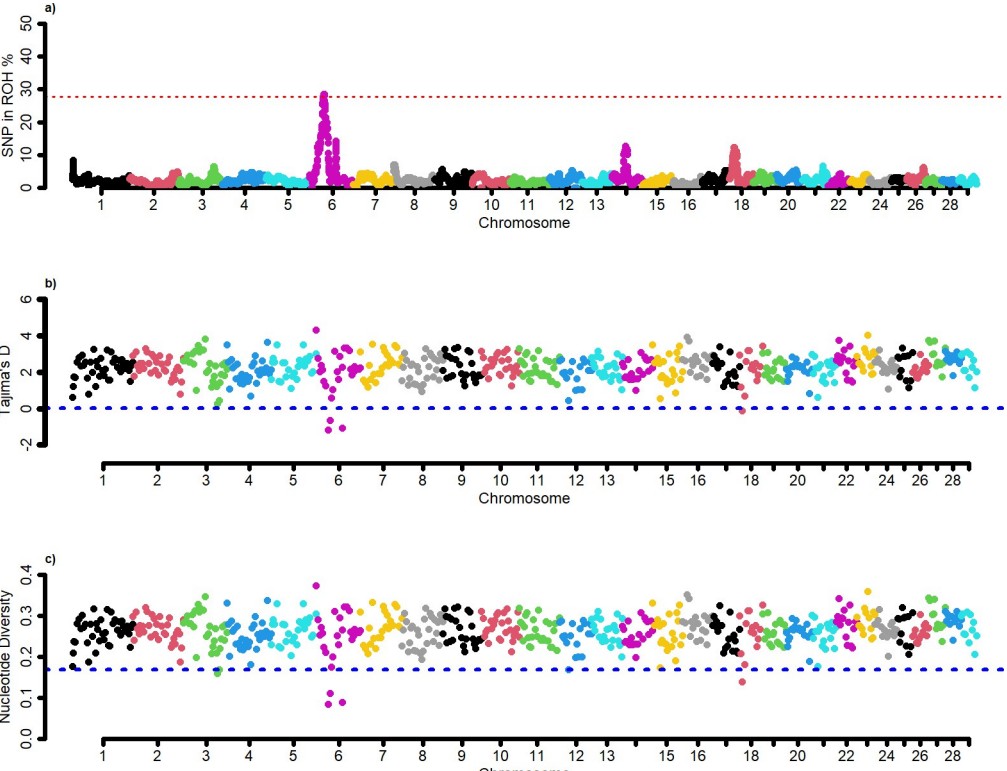

**Fig 2. Detection of selection signatures through intra-population statistics in Sardo Negro breed.** a) Manhattan plot of the SNP frequency within runs of homozygosity b) Manhattan plot of nucleotide diversity. c) Manhattan plot of Tajima´s D values.

**Table 2. Runs of Homozygosity (ROH) island, genes, and GO terms in Sardo Negro cattle.**

| No SNP | ROH island | Genes | GO terms |
|---|---|---|---|
| 38 | 6:35.78–42.51 Mb | *CCSER1 MMRN1 SNCA FAM13A HERC3 NAP1L5 PIGY PYURF HERC5 PPM1K ABCG2 PKD2 SPP1 IBSP MEPE TRNAA-CGC LAP3 MED28 FAM184B NCAPG DCAF16 LCORL SLIT2 PACRGL KCNIP4* | DNA-binding transcription factor, transferase, cytokine, ATP-binding cassette (ABC) transporter, ubiquitin-protein ligase, scaffold/adaptor protein, ATP-binding cassette (ABC) transporter, membrane trafficking regulatory protein, protein phosphatase, ion channel, cytokine, extracellular matrix structural protein, general transcription factor, metalloprotease, calmodulin-related, ubiquitin-protein ligase |

Only one candidate region overlapped between all four methods, ROH, Nucleotide diversity, Tajima's D, and $F_{ST}$ (34.8.47–43.2 MB). That region involves the *CCSER1*, *MMRN1*, *SNCA*, *FAM13A*, *HERC3*, *NAP1L5*, *PIGY*, *PYURF*, *HERC5*, *PPM1K*, *ABCG2*, *PKD2*, *SPP1*, *IBSP*, *MEPE*, *TRNAA-CGC*, *LAP3*, *MED28*, *FAM184B*, *NCAPG*, *DCAF16*, *LCORL*, *SLIT2*, *PACRGL*, and *KCNIP4* genes and QTLs related to pelvic area, body weight, marbling score, milk alpha-S1-casein percentage, and milk C18 index [19].

## Discussion

Several researchers have explored the selection signatures of cattle, pigs, sheep, and horses [20]. In the present study, we aimed to identify signatures of selection in the Sardo Negro Zebu breed using bovine 50K SNP chip data. Many studies have used the bovine 50K SNP chip to detect selection signatures in various cattle breeds [21,22]. However, compared to *Bos taurus* cattle breeds, the number of studies in *Bos indicus* cattle has been limited [3].

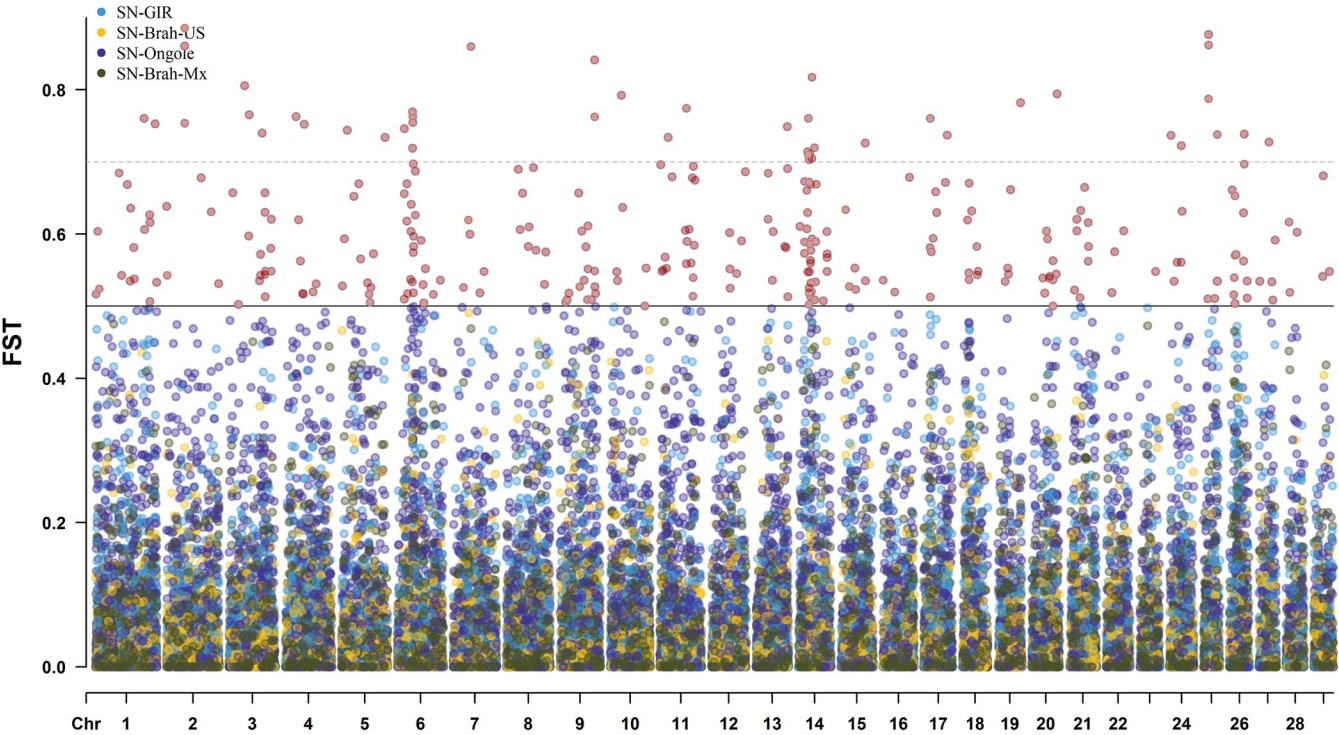

**Fig 3. Overlapped Genome-wide distribution of selection signatures detected by $F_{ST}$.** Sardo Negro (SN) were compared with Gir, Brahman [American (Brah-US) and Mexican (Brah-Mx)] samples, and Ongole zebu breeds.

Four complementary methodologies were applied to uncover selection signatures in the studied population, which should improve detection accuracy and eliminate unknown bias [20,23]. In cattle, using a low-density SNP microarray tends to overestimate the number of ROH $\leq$ 4 Mb. In contrast, using a high-density chip could lead to an underestimation of the number of long ROH (i.e., $\geq$8 Mb). However, De Roos et al. [24] suggested that 50k SNPs yield reasonable parameters when studying one breed. Using the ROH approach, we detected a selection sweep based on identifying genomic regions with reduced variation relative to the genome average, as Broman and Weber [25] suggested for the first time.

We found only one selective sweep on BTA6 by ROH. Maiorano et al. [26] also showed the presence of a selection signature on BTA6 for the Gir dairy population. These authors mentioned that *LCORL* and *NCAPG* genes on BTA6 were found to be associated with growth, meat, and carcass traits. This similarity can be explained because the founder breeds of the Sardo Negro breed are the Gir and Indubrasil [1]. In other words, the selective sweep detected on the BTA6 could be due to selection history in their founder breeds. The length of the ROH is considered a helpful time indicator of its associated inbreeding event. Long ROHs are related to recent inbreeding events in a breed's history, whereas short ROHs indicate a more ancient event [27].

The positive selection signals around this region are further confirmed by significantly lower values of Tajima's D. The negative value shows the fixation of rare alleles. This is indicated by a smaller value (more negative) of Tajima's D, suggesting an abundance of rare alleles and positive selection [28]. Positive values show the balancing selection, which reflects an abundance of intermediate allele frequencies. The zero D value is the indication of neutral variation.

Many studies relate that region on BTA6 to beef cattle's feed intake, gain, meat, and carcass traits [29,30]. Also, other genes found on this selective sweep, like *LAP3*, *MED28*, *SLIT2*, and *IBSP*, have been previously reported to be associated with carcass traits, body measurements, growth traits, and bone weight [31–33]. Niu et al. [34] reported that the region on BTA6, found like a selective sweep in our study, provides valuable information for understanding the genetic basis of body stature in beef cattle.

Dominguez Viveros et al. [35] characterized the growth of five zebu breeds including Sardo Negro. Sardo Negro had the highest values in the parameter related to mature weight, (871 kg in males, and 615 kg in females). The Gir and Indubrasil breeds, considered as the ancestor breeds of Sardo Negro animals, had lower values in mature weight (Indbrasil:744kg and 506 kg; Gir:695 kg and 527 kg) than Sardo Negro [36].

Comparing the ROHs of the Sardo Negro Zebu breed with the ROHs observed in the samples of other Zebu breeds like Gir, Brahman, and Ongole (S5–S8 Figs). None of the reference populations exhibit ROHs similar to those found in the Sardo Negro population. Additionally, the ROHs in BTA6, identified as selection signatures, were not observed in any other reference population.

Other signals of diversifying selection were found comparing the Sardo Negro Zebu breed and other Zebu breeds like Gir, Brahman, and Ongole. We only compared Sardo Negro with Zebu breeds because of the known genomic divergence of zebu and taurine subspecies [37]. Nine SNPs above the threshold of the top 0.1% $F_{ST}$ value were commonly detected among the comparison between Sardo Negro and Gir, Brahman, and Ongole Zebu breeds. The top SNPs above the $F_{ST}$ threshold between Sardo Negro and Mexican Brahman only coincide by region, not by SNP, with the other $F_{ST}$ comparisons, because of the difference in the used SNP chip. The candidate divergent regions detected (± 250kb of each $F_{ST}$ peak) in the present study contain several QTLs for live weight traits. A region mapped on BTA8 (8:93.4–93.9 Mb) had been related to selenoproteins that play crucial roles in the regulation of oxidative stress and calcium

homeostasis in striated muscle regarding the maintenance of functions and disease progression [37,38].

The window on BTA11 (11:99.2–99.7 Mb), which includes the *OLFML2A*, *WDR38*, *RPL35*, *ARPC5L*, and *GOLGA1* genes, which have been associated with the fat percentage in milk. The diversifying selection region on BTA14 harbors a QTL previously reported to be involved in milk and fat protein percentage. Soares et al. [39] identified a relation between the *CHD7* and *RAB2A* genes with fertility traits in Brahman cattle. *CHD7* has been reported to be associated with age at first corpus luteum and scrotal circumference.

The selection signatures found in this study may be affected by several factors, including sample size, the SNP density, and ascertainment bias in the genomic data [23]. However, using two approaches strengthens the identification of the selection signatures, highlighting the 35.78–42.51 Mb region on BTA6.

## Conclusions

Using two complementary statistical approaches (ROH and $F_{ST}$) facilitated the broader spectrum of detection of selection signatures. The selective signals identified in this study reflected the direction of the selection pressure that primarily involves the increase of live weight traits in the Sardo Negro cattle population. The candidate region 35.78–42.51 Mb on BTA6 might be related to body weight identified in Sardo Negro in Mexico.

## Supporting information

**S1 Fig. Genome-wide distribution of selection signatures detected by $F_{ST}$ between Sardo Negro and Gir.** The horizontal line represents the threshold level of 0.1%.
(TIF)

**S2 Fig. Genome-wide distribution of selection signatures detected by $F_{ST}$ between Sardo Negro and Brahman (American sample).** The horizontal line represents the threshold level of 0.1%.
(TIF)

**S3 Fig. Genome-wide distribution of selection signatures detected by $F_{ST}$ between Sardo Negro and Ongole.** The horizontal line represents the threshold level of 0.1%.
(TIF)

**S4 Fig. Genome-wide distribution of selection signatures detected by $F_{ST}$ between Sardo Negro and Brahman (Mexican sample).** The horizontal line represents the threshold level of 0.1%.
(TIF)

**S5 Fig. Manhattan plot of the SNP frequency within runs of homozygosity by autosome in a Mexican Brahman zebu cattle population sample.** The horizontal dashed line represents the threshold level of 0.1%.
(TIF)

**S6 Fig. Manhattan plot of the SNP frequency within runs of homozygosity by autosome in an American Brahman zebu cattle population sample.** The horizontal dashed line represents the threshold level of 0.1%.
(TIF)

**S7 Fig. Manhattan plot of the SNP frequency within runs of homozygosity by autosome in a Gir zebu cattle population sample.** The horizontal dashed line represents the threshold

level of 0.1%.
(TIF)

**S8 Fig. Manhattan plot of the SNP frequency within runs of homozygosity by autosome in an Ongole zebu cattle population sample.** The horizontal dashed line represents the threshold level of 0.1%.
(TIF)

**S1 Table. SNPs above 0.1% of the highest $F_{ST}$ values comparing Sardo Negro with Gir, Brahman, and Ongole.**
(XLSX)

**S2 Table. Genomic regions under divergent selection identified by FST approach and associated candidate genes, QTLs and GO terms.**
(XLSX)

## Acknowledgments

We thank the Asociación Mexicana de Criadores de Cebu (AMCC) for providing information for the present study. We also are grateful to DGAPA-UNAM PAPIIT-IA202323.

## Author Contributions

**Conceptualization:** Joel Domínguez Viveros.

**Data curation:** Joel Domínguez Viveros, Antonio Reyes Ceron.

**Formal analysis:** Ricardo Martínez-Rocha, Jorge Hidalgo.

**Investigation:** Rodolfo Ramírez Valverde, Joel Domínguez Viveros.

**Methodology:** Rafael Núñez Domínguez, Jorge Hidalgo.

**Supervision:** Rafael Núñez Domínguez.

**Writing – original draft:** Victor Isaias Garduño López, Ricardo Martínez-Rocha, Rodolfo Ramírez Valverde, Joel Domínguez Viveros.

**Writing – review & editing:** Victor Isaias Garduño López, Rafael Núñez Domínguez, Rodolfo Ramírez Valverde, Jorge Hidalgo.

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
