## [Decision Letter · Decision Letter 0]

12 Aug 2024

PONE-D-24-17824Genome-wide scan for selection signatures in Mexican Sardo Negro Zebu cattlePLOS ONE

Dear Dr. Martínez-Rocha,

Thank you for submitting your manuscript to PLOS ONE. After careful consideration, we feel that it has merit but does not fully meet PLOS ONE’s publication criteria as it currently stands. Therefore, we invite you to submit a revised version of the manuscript that addresses the points raised during the review process.

We look forward to receiving your revised manuscript.

Kind regards,

Amod Kumar, Ph.D

Academic Editor

PLOS ONE

Journal Requirements:

3. In the online submission form, you indicated that [Data available on request from the author]. 

Reviewers' comments:

Reviewer's Responses to Questions

**Comments to the Author**

1. Is the manuscript technically sound, and do the data support the conclusions?

Reviewer #1: Partly

Reviewer #2: Yes

2. Has the statistical analysis been performed appropriately and rigorously? 

Reviewer #1: I Don't Know

Reviewer #2: Yes

3. Have the authors made all data underlying the findings in their manuscript fully available?

Reviewer #1: No

Reviewer #2: No

4. Is the manuscript presented in an intelligible fashion and written in standard English?

Reviewer #1: Yes

Reviewer #2: Yes

5. Review Comments to the Author

Reviewer #1: Manuscript_ PONE-D-24-17824

Dear editors,

The manuscript: “Genome-wide scan for selection signatures in Mexican Sardo Negro zebu cattle” could provide an additional interesting piece of information on genomic selection effects in a regionally selected zebu cattle population.

As this population had been selected since the fifties of the last century on pedigree basis, it provides a valuable resource for the family based inheritance on inbreeding. In the context of this report, the number of founder bulls and the proportional representation in the current population would be extremely interesting.

Unfortunately, the authors did not include the information available for their sample in this report yet.

Which effective family size is represented by each of the different Zebu founder genotypes in the pedigree data.

On which criteria had the samples for this study been selected? How many different bull families are included and how do they represent progeny of the different founder bull genotypes of the SN in the sample selected for the SNP analysis?

These data are prerequisite to evaluate the ROH and Fst data of this study.

What are the phenotypic data changes for the mentioned selection criteria (morphology, body conformation, production) within the populations and compared to the reference populations / samples that had been investigated? What phenotype data for the live body weight are representative for the different samples?

Are in the reference samples for the Zebu founder populations ROH for the same or different alleles in the respective regions? How many different SNPs in and between the Zebu-populations could be assigned for the ROH / Fst regions highlighted interesting on chromosomes 6, 8, 11 and 14?

The manuscript should be revised including the requested information bevor considering re-review.

Reviewer #2: 1. Kindly provide the rationale behind selection of only two methodologies (ROH and Fst) for this study.

2. The SN population was genotyped using Axiom platform, whereas others on the Illumina platform. How could you identify the common genetic markers for Fst method?

3. Could you explain that both SNP arrays (Axiom and Illumina) developed using the same genome assembly or not?

4. How could you decide the parameters for ROH detection? Please explain.

5. Have you identified the common genes between these two methods?

6. Kindly include nucleotide diversity method for selection signature identification, ans use DCMS methods for identification of common signals.

6. PLOS authors have the option to publish the peer review history of their article (what does this mean?). If published, this will include your full peer review and any attached files.

Reviewer #1: No

Reviewer #2: No

---

## [Author Response · Author response to Decision Letter 0]

21 Aug 2024

We want to thank the two reviewers for their comments, which allowed us to improve the quality of the paper. The manuscript was modified according to the comments, and all changes will be found in a marked-up copy of our manuscript.

Response to Reviewer 1 Comments

Reviewer: 1

The manuscript: “Genome-wide scan for selection signatures in Mexican Sardo Negro zebu cattle” could provide an additional interesting piece of information on genomic selection effects in a regionally selected zebu cattle population.

As this population had been selected since the fifties of the last century on pedigree basis, it provides a valuable resource for the family based inheritance on inbreeding. 

In the context of this report, the number of founder bulls and the proportional representation in the current population would be extremely interesting.

Unfortunately, the authors did not include the information available for their sample in this report yet. Which effective family size is represented by each of the different Zebu founder genotypes in the pedigree data.

Response 1. Thanks for the comments. The information about pedigree-based and genome-based (the same sample of the recent work) diversity already have been evaluated in previous works. This information can be found at doi.org/10.1590/0103-8478cr20210116, and doi.org/10.1016/j.livsci.2023.105267 respectively.

On which criteria had the samples for this study been selected? How many different bull families are included and how do they represent progeny of the different founder bull genotypes of the SN in the sample selected for the SNP analysis?

These data are prerequisite to evaluate the ROH and Fst data of this study.

Response 2. Our criteria for selecting animals in the study was that the study samples were chosen randomly, and only included animals with no parent-offspring relationship. Only animals were considered if they were purebred based on pedigree and farmer interviews.

What are the phenotypic data changes for the mentioned selection criteria (morphology, body conformation, production) within the populations and compared to the reference populations / samples that had been investigated? What phenotype data for the live body weight are representative for the different samples?

Response 3. Thank you, we aggregated more information in the manuscript related to phenotypic information of the Sardo Negro breed. This information is already detailed in https://doi.org/10.1016/j.livsci.2020.104303.

Are in the reference samples for the Zebu founder populations ROH for the same or different alleles in the respective regions?

Response 4. We included ROH identified in the reference populations used in the supplementary data (S5-S8 Figs).

 How many different SNPs in and between the Zebu-populations could be assigned for the ROH / Fst regions highlighted interesting on chromosomes 6, 8, 11 and 14?

Response 5. The information was already included in the supplementary data (S1-S2 Table).

Reviewer #2: 1. Kindly provide the rationale behind selection of only two methodologies (ROH and Fst) for this study

Response 1. Thanks for your feedback. We aimed to select at least one statistic within a population and one statistic across populations to visually represent signs of selection in these two ways. However, we also considered the sixth point, and we have already aggregated DCMS methods.

2. The SN population was genotyped using Axiom platform, whereas others on the Illumina platform. How could you identify the common genetic markers for Fst method?

Response 2. Thank you. We didn’t clarify this point previously. We identified the common genetic markers mapping SNPs to a common assembly using SNPchiMp database (doi:10.1186/1471-2164-15-123).

3. Could you explain that both SNP arrays (Axiom and Illumina) developed using the same genome assembly or not?

Axiom bovine SNP array is aligned to the Bos taurus genome assembly ARS-UCD1.2, and Illumina was developed using UMD3.1 assembly. However, we used SNPchiMp to map the SNPs from both arrays to a common reference genome. We carefully check the harmonization of alleles ensuring that the alleles are consistent between arrays. If the SNPs had different representations of alleles or different strand orientations, we used --flip and –flip-scan functions of PLINK.

4. How could you decide the parameters for ROH detection? Please explain.

We decide on the minimum length of ROH of 1MB and the minimum number of 20 consecutive homozygous SNPs because, with a lower SNP density, we might detect shorter ROHs. Is known that shorter ROHs are related to ancient events in the selection signatures.

We opted to set the maximum gap at 1 Mb since it's customary to permit a maximum gap of 1-2 megabases (Mb) between SNPs as a starting point. This parameter specifies the largest acceptable distance between consecutive SNPs for them to be categorized as part of the same ROH 

The maximum number of missing allowed genotypes was set to 1 because it results in 5% of Missing Data; a common threshold is to allow up to 5% of missing genotypes within a given ROH.

5. Have you identified the common genes between these two methods?

After considering the sixth point, the common genes between the four methods is only on BTA6. That region involves the CCSER1, MMRN1, SNCA, FAM13A, HERC3, NAP1L5, PIGY, PYURF, HERC5, PPM1K, ABCG2, PKD2, SPP1, IBSP, MEPE, TRNAA-CGC, LAP3, MED28, FAM184B, NCAPG, DCAF16, LCORL, SLIT2, PACRGL, and KCNIP4 genes.

6. Kindly include nucleotide diversity method for selection signature identification, and use DCMS methods for identification of common signals.

Done

---

## [Decision Letter · Decision Letter 1]

8 Oct 2024

Genome-wide scan for selection signatures in Mexican Sardo Negro Zebu cattle

PONE-D-24-17824R1

Dear Dr. Martinez-Rocha,

We’re pleased to inform you that your manuscript has been judged scientifically suitable for publication and will be formally accepted for publication once it meets all outstanding technical requirements.

Kind regards,

Amod Kumar, Ph.D

Academic Editor

PLOS ONE

Additional Editor Comments (optional):

Congratulations!!!

Authors have addressed all the queries raised by the reviewers.

Best Regards

Reviewers' comments:

Reviewer's Responses to Questions

**Comments to the Author**

Reviewer #1: All comments have been addressed

Reviewer #2: All comments have been addressed

2. Is the manuscript technically sound, and do the data support the conclusions?

Reviewer #1: Yes

Reviewer #2: Yes

3. Has the statistical analysis been performed appropriately and rigorously? 

Reviewer #1: Yes

Reviewer #2: Yes

4. Have the authors made all data underlying the findings in their manuscript fully available?

Reviewer #1: Yes

Reviewer #2: (No Response)

5. Is the manuscript presented in an intelligible fashion and written in standard English?

Reviewer #1: Yes

Reviewer #2: Yes

6. Review Comments to the Author

As suggested by reviewer 1, please included references in the manuscript.

Reviewer #1: Please add the two references regarding the pedigree data of the Mexican Sardo Negro Zebu cattle to this manuscript to allow forin depth information of the readers.

Reviewer #2: (No Response)

7. PLOS authors have the option to publish the peer review history of their article (what does this mean?). If published, this will include your full peer review and any attached files.

Reviewer #1: **Yes: **Sabine Klein, Friedrich-Loeffler-Institut, Institute of Farm Animal Genetics Mariensee

Reviewer #2: No

---

## [Editor Report · Acceptance letter]

31 Oct 2024

PONE-D-24-17824R1 

PLOS ONE

Dear Dr. Martínez-Rocha, 

I'm pleased to inform you that your manuscript has been deemed suitable for publication in PLOS ONE. Congratulations! Your manuscript is now being handed over to our production team.

Kind regards, 

on behalf of

Dr. Amod Kumar 

Academic Editor

PLOS ONE